# Age of initiation of hookah use among young adults: Findings from the Population Assessment of Tobacco and Health (PATH) study, 2013–2017

Adriana Pérez [1,2]*, Arnold E. Kuk[2], Meagan A. Bluestein[2], Baojiang Chen[1,2], Kymberle L. Sterling[3], Melissa B. Harrell[2,4]

1 Department of Biostatistics and Data Science, The University of Texas Health Science Center at Houston (UTHealth), School of Public Health, Austin, Texas, United States of America, 2 Michael & Susan Dell Center for Healthy Living, The University of Texas Health Science Center at Houston (UTHealth), School of Public Health, Austin, Texas, United States of America, 3 Department of Health Promotion and Behavioral Sciences, The University of Texas Health Science Center at Houston (UTHealth), School of Public Health, Dallas, Texas, United States of America, 4 Department of Epidemiology, Human Genetics and Environmental Sciences, The University of Texas Health Science Center at Houston (UTHealth), School of Public Health in Austin Campus, Austin, Texas, United States of America

* adriana.perez@uth.tmc.edu

**Data Availability Statement:** All the data from waves 1-4 are available from the Population Assessment of Tobacco and Health (PATH) Study

## Abstract

### Objective

To prospectively estimate the age of initiation of ever, past 30-day, and fairly regular hookah use among young adults (ages 18–24) overall, by sex, by race/ethnicity, and to explore the association of prior use of other tobacco products with these hookah use behaviors.

### Methods

Secondary data analyses of the first four waves (2013–2017) of the PATH study, a nationally representative longitudinal cohort study of US young adults. Young adult never hookah users at the first wave of adult participation in PATH waves 1–3 (2013–2016) were followed-up into waves 2–4 (2014–2017) to estimate the age of initiation of three outcomes: (i) ever use, (ii) past 30-day use, and (iii) fairly regular hookah use. Weighted interval-censoring Cox proportional hazards regression models were used to examine the differences in the estimated age of initiation by sex and by race/ethnicity while controlling for the total number of other tobacco products ever used at participants' first wave of PATH participation. In addition, to examine if prior use of other tobacco products was associated with the age of hookah initiation behaviors, six additional Cox models are reported for each hookah initiation behaviors.

### Results

The largest increase in hookah use occurred between ages 18 and 19: 5.8% for ever use and 2.7% for past 30-day hookah use. By age 21, 10.5%, 4.7% and 1.2% reported initiation of ever, past 30-day and fairly regular hookah use, respectively. There were statistically significance differences in the age of initiation of hookah use behaviors by race/ethnicity.

[United States] Restricted-Use Files. Inter-university Consortium for Political and Social Research [distributor], 2020-06-24. https://doi.org/10.3886/ICPSR36231.v27. Researchers can apply for access to the restricted-use datasets from the Inter-university Consortium for Political and Social Research (ICPSR) at the University of Michigan. To access data in the Virtual Data Enclave (VDE), a Restricted Data Use Agreement (RDUA) must be established between the University of Michigan and the researcher's institution. Data are provided via ICPSR's VDE. For further information, please reference the VDE Guide to learn about the application process, about using the VDE, and how to request disclosure review of VDE output located here: https://www.icpsr.umich.edu/web/pages/NAHDAP/vde/index.html Obtaining results using the restricted-use datasets requires a disclosure process with protocols set by ICPSR. When a researcher logs on to the VDE, a virtual machine is launched on the researcher's own desktop but operates from a server at ICPSR. The virtual machine is isolated from the researcher's physical desktop computer - users cannot download or upload files or parts of files from or to the VDE; print VDE contents to a printer; or email, copy, or otherwise move files in or out of the VDE computing environment, either accidentally or intentionally. Results are only disclosed by ICPSR after programs have been checked for accuracy and results have been replicated.

**Funding:** Research reported in this publication was supported by grant number [CA2342051] from the National Cancer Institute (NCI) and the FDA Center for Tobacco Products (CTP) to Dr. Adriana Pérez. The content is solely the responsibility of the authors and does not necessarily represent the official views of the National Institutes of Health (NIH) or the Food and Drug Administration (FDA).

**Competing interests:** The authors have no conflicts of interest to disclose except Dr. Harrell is a consultant in litigation involving the vaping industry. This does not alter our adherence to PLOS ONE policies on sharing data and materials.

## Conclusion

Educational interventions should target young adults before the age of 21, focusing efforts specifically on males, non-Hispanic Blacks and Hispanics, to stall initiation and progression of hookah use behaviors.

## Introduction

The popularity of hookah use can be contributed to many factors, such as misperceptions that it is less harmful and not as addictive as cigarettes [1–3]. However, a systematic review in 2019 indicated that a chemical analysis of hookah charcoal, which is used to heat the shisha or hookah tobacco, identified 7 carcinogens, 39 central nervous system depressants, and 31 respiratory irritants, which are associated with potential health risks [2–4]. Studies suggest that hookah use exposes users to chemicals inducing nicotine dependence [2,3,5,6] and results in higher risk of oral health deterioration, gastro-esophageal reflux disease, low birth weight, cancers, cardiovascular diseases, and pulmonary diseases [2]. Moreover, studies further suggest that hookah smoking can cause acute carbon monoxide poisoning that often requires hospitalization [7–9] and serves as a gateway to cigarette use [10–12].

Hookah use has caught the attention of researchers and health professionals because of its rising popularity among young adults and the health risks associated with its use [2,4,13,14]. Between 2009–2010 and 2013–2014, the prevalence of past 30-day hookah use among young adults (aged 18–24) increased from 7.8% to 18.2% and ever hookah use increased from 28.6% to 44.4% [15–17]. A different study found that among 18–24 year old never hookah users in 2013–2014, 14.1% initiated ever use, 6.1% initiated past 30-day use, and 0.2% initiated frequent hookah use (i.e. use on at least 20 days in the past 30-days) one year later [18].

Given the popularity of hookah use among young adults and its negative health effects, understanding the age that young adults initiate use will be useful to inform prevention intervention efforts. In this study, we prospectively estimated the distributions of the age of initiation of hookah use among U.S. young adults, aged 18–24, who were never hookah users at their first wave of participation of the Population Assessment of Tobacco and Health (PATH) adult study. Three behavioral outcomes are reported: age of initiation of (i) ever hookah use, (ii) past 30-day hookah use, and (iii) fairly regular hookah use. The current study goes beyond prior research that has provided prevalence (yes/no) of hookah use [15,16,19,20] by analyzing age with each yes/no initiation outcome. Prior studies provide evidence that hookah use among adults (aged 18+) varies by demographic factors such as sex and race/ethnicity. We therefore explore differences in the age of initiation of each hookah use behavior by sex and by race/ethnicity [15,19,21–23]. A PATH study from 2013–2016 examined the role of other tobacco product use correlates on the initiation of hookah, which found that past 30-day hookah use was associated with ever e-cigarette use and ever cigar products use [24]. For this reason, we explored if prior use of other tobacco products was associated with the age of initiation of hookah use behaviors with the number of other tobacco products ever used as control variables.

## Methods

### Study design and participants

The PATH study is an ongoing nationally representative longitudinal cohort study of U.S. youth (ages 12–17) and adults (ages 18+) conducted annually since wave 1 2013–2014, collecting information from the participants about their demographics, tobacco use behaviors,

substance use behaviors, beliefs and attitudes toward tobacco use, and tobacco-related health outcomes. Details of the study design have been published previously [25].

This study conducted secondary analyses using the PATH restricted-use adult datasets among young adult (18–24 years old) never users of hookah in waves 1–3 with behavioral outcomes followed-up in waves 2–4 (wave 2: 2014–2015, wave 3: 2015–2016, wave 4: 2016–2017). The wave 1 young adult (18–24 years old) sample included 9,110 participants, of which, 4,034 (N = 16,991,951) were never hookah users. Youth who turned 18 at waves 2 and 3 were invited to participate in the adult study, and there were 1,276 (N = 2,831,511) never hookah users at wave 2 and 1,321 (N = 2,846,135) never hookah users at wave 3. This resulted in a sample size of 6,631, representing 22,669,597 young adult never hookah users at their first wave of participation in PATH (waves 1–3) included in our study. IRB approval for this study was obtained from the Committee for the Protection of Human Subjects at the University of Texas Health Science Center at Houston with number HSC-SPH-17-0368 and ICPSR approved our restricted-use access to PATH data. The original investigators of the PATH study obtained written informed consent for adult participants.

## Outcomes

**Ever use.** At wave 1, ever use of hookah was measured with the question: "Have you ever smoked tobacco in a hookah, even one or two puffs?". At waves 2–4, participants who were never user of hookah at the previous wave were asked the question: "In the past 12 months, have you smoked tobacco in a hookah, even one or two puffs?". Response options included "yes", "no", and "don't know". Participants who answered "yes" were considered ever hookah users.

**Past 30-day use.** At each wave, participants who had ever used hookah were asked: "In the past 30 days, have you smoked hookah, even one or two puffs?". Response options included "yes", "no", and "don't know". Participants who answered "yes" were considered past 30-day hookah users.

**Fairly regular use.** At each wave, participants who had ever used hookah were asked: "Have you ever smoked hookah fairly regularly?". Response options included "yes", "no", and "don't know". Participants who answered "yes" were considered fairly regular hookah users.

Responses from participants who refused to answer or who responded "don't know" for ever use, past 30-day or fairly regular hookah use were considered missing.

**Other tobacco product use.** At each wave, ever use was measured for the following tobacco products: cigarettes, e-cigarettes, traditional cigars, cigarillos, filtered cigars, and smokeless tobacco. To control for the effect of other tobacco product use, a variable was created to represent the total number of other tobacco products ever used (0–6) at participants' first wave of PATH participation (waves 1–3). The total number of other tobacco products ever used was then categorized as 0, 1, and 2+ other tobacco products ever used. In addition, to evaluate the association of each specific tobacco product prior to hookah use behaviors, dichotomous variables (yes/no) were created to represent ever use of each tobacco product (cigarettes, e-cigarettes, traditional cigars, cigarillos, filtered cigars, and smokeless tobacco) prior to initiation of hookah use behaviors for each outcome (6 tobacco products*3 outcomes = 18 variables).

**Demographic variables.** PATH provided a variable for sex categorizing participants as either male or female. For a few respondents who did not provide their sex, PATH imputed their sex based on the household screener data at wave 1, but not at waves 2–4. Four categories were used to measure participant race in PATH: White race alone, Black race alone, Asian race alone, and other race (including multi-racial). Participants' ethnicity was categorized as either

Hispanic or Non-Hispanic. Following the Surgeon General's report [13], we combined race and ethnicity into four categories: Non-Hispanic White, Non-Hispanic Black, Hispanic, and Non-Hispanic Other (Asian, multi-race, and etc.). Participants who reported Hispanic ethnicity and missing race were coded as Hispanic; participants who reported White race alone and missing ethnicity were coded as Non-Hispanic White; participants who reported Black race alone and missing ethnicity were coded as Non-Hispanic Black; participants who reported Asian/other race alone and missing ethnicity were coded as Non-Hispanic other.

**Age of initiation of hookah.** The PATH study does not provide participants' birth dates and it is unlikely that the participants' can identify their exact date of hookah initiation. However, PATH provided a derived variable for participant age in years at each wave (waves 1–4). In addition, PATH provided a variable representing the number of weeks between waves of participation, which was calculated by assigning the calendar week of the year (0–52) to the date that the survey was conducted. Age of initiation of ever hookah use, past 30-day hookah use, and fairly regular hookah use was estimated by adding the participants' age at the first wave (waves 1–3) in which they were never hookah users to the number of weeks between relevant subsequent waves, 2014–2017 (waves 2–4), in which the participants first reported initiating each hookah use behaviors. Participant age was calculated in weeks and added to the number of weeks between waves of study participation to provide a more precise estimate of participant age rather than using age in years at each wave. For all participants, the lower age bound was the age at the last wave when a participant reported non-use of each hookah outcome. For those who become users, the upper age bound was calculated by adding the lower age bound and the number of weeks between the last wave a participant reported non-use and the first wave they reported hookah initiation. For those who did not report initiating hookah use, the upper age bound was censored.

## Statistical analysis

There was very little missingness in PATH, and missing values are reported. All statistical analyses were conducted using SAS version 9.4. PATH uses a complex survey design, requiring the use of sampling weights and 100 balanced repeated replicate weights with Fay's correction factor of 0.3 [25,26]. Weighted summary statistics for means and proportions are provided for young adult never hookah users at their first wave of study participation. The age of initiation of ever, past 30-day, and fairly regular hookah use was estimated using weighted interval-censoring survival analysis. The hazard function (and its 95% confidence intervals) for each outcome was estimated using the Turnbull non-parametric estimator [27], and are reported as cumulative incidence in percentages, which are presented in figures. Weighted interval-censoring Cox proportional hazards regression models [28] were used to examine the differences in the estimated age of initiation by sex and by race/ethnicity while controlling for the total number of other tobacco products ever used at participants' first wave of participation. If sex or race/ethnicity were statistically associated with the age of each hookah initiation behaviors, weighted interval-censoring survival analyses were conducted stratified by sex and by race/ethnicity.

In addition, to examine if prior use of each other tobacco product was associated with the age of hookah initiation behaviors, six additional crude weighted interval-censoring Cox models are reported for each hookah initiation behavior that account for ever use of each of the six other tobacco products using separate variables. In addition, weighted interval-censoring multivariable Cox models including sex, race/ethnicity, and each of the six other tobacco products ever used prior to hookah use as covariates for the age of initiation of (i) ever, (ii) past 30-day, and (iii) fairly regular hookah use are included in order to examine the contribution of each tobacco product separately.

## Results

Demographic characteristics of PATH young adult never hookah users at their first wave of PATH adult participation are presented in Table 1. Their average age was 20 years old, with almost 75% of them entering the study at wave 1 (2013–2014), 51.2% were males, 53.3% were non-Hispanic White, 28.5% had ever used cigarettes prior to initiation of ever hookah use. The proportions of ever use of the other 5 tobacco products prior to hookah initiation are reported in Table 1. Descriptive statistics for ever use of other tobacco products prior to initiation of past 30-day and fairly regular hookah use are provided in S1 Table.

Table 2 shows the distribution of the age of initiating hookah use behaviors for U.S. young adults who were never hookah users at their first wave PATH adult participation. By age 21, it is estimated that 10.5% of young adults initiated ever hookah use, 4.7% initiated past 30-day hookah use, and 1.2% initiated fairly regular hookah use. By age 25, it is estimated that 20.4% of young adults initiated ever hookah use, 10.2% initiated past 30-day hookah use, and 2.1%

**Table 1. Demographic characteristics of PATH¥ USA young adult (aged 18–24) never hookah users at their first wave of adult participation (2013–2016).**

| | | | N = 6,631; N = 22,669,597 | |
|---|---|---|---|---|
| | | | Unweighted n (N) | Weighted % (SE) |
| **Wave of Entry Into PATH** | Wave 1 (2013–2014) | | 4,034 (16,991,951) | 74.9% (0.33) |
| | Wave 2 (2014–2015) | | 1,276 (2,831,511) | 12.5% (0.23) |
| | Wave 3 (2015–2016) | | 1,321 (2,846,135) | 12.6% (0.27) |
| **Age at entry into study (SE)** | Weighted mean (SE) | | 20.17 (0.04) | |
| **Sex** | Female | | 11,605,270 | 51.2% (0.47) |
| | Male | | 11,052,716 | 48.8% (0.47) |
| | Missing | | 11,610 | |
| **Race/Ethnicity** | Non-Hispanic White | | 12,078,728 | 53.3% (1.09) |
| | Hispanic | | 4,548,123 | 20.1% (0.66) |
| | Non-Hispanic Black | | 3,480,515 | 15.4% (0.56) |
| | Non-Hispanic Other* | | 2,562,231 | 11.3% (0.82) |
| **Total number of other tobacco product ever used** | 0 other tobacco products | | 3,396 (13,702,650) | 60.5% (1.05) |
| | 1 other tobacco product | | 1,096 (3,513,346) | 15.5% (0.58) |
| | 2+ other tobacco products | | 2,139 (5,453,601) | 24.1% (0.79) |
| **Prior tobacco product use** | Cigarette | No | 4,319 (16,206,614) | 71.5% (0.94) |
| | | Yes | 2,312 (6,462,983) | 28.5% (0.94) |
| | E-cigarette | No | 4,704 (17,785,558) | 78.5% (0.78) |
| | | Yes | 1,927 (4,884,038) | 21.5% (0.78) |
| | Smokeless Tobacco | No | 6,037 (21,105,205) | 93.1% (0.40) |
| | | Yes | 594 (1,564,392) | 6.9% (0.40) |
| | Traditional Cigars | No | 5,824 (20,345,760) | 89.7% (0.52) |
| | | Yes | 807 (2,323,836) | 10.3% (0.52) |
| | Filtered Cigars | No | 5,969 (20,964,388) | 92.5% (0.40) |
| | | Yes | 662 (1,705,209) | 7.5% (0.40) |
| | Cigarillo | No | 5,065 (18,487,849) | 81.6% (0.64) |
| | | Yes | 1,566 (4,181,748) | 18.4% (0.64) |

*Non-Hispanic Other includes Asian, multi-race, and etc.

¥ PATH restricted file received disclosure to publish: April 14, 2020, July 23, 2020, and March 29, 2021. United States Department of Health and Human Services. National Institutes of Health. National Institute on Drug Abuse, and United States Department of Health and Human Services. Food and Drug Administration. Center for Tobacco Products. Population Assessment of Tobacco and Health (PATH) Study [United States] Restricted-Use Files. ICPSR36231-v13.AnnArbor, MI: Inter-university Consortium for Political and Social Research [distributor], November 5, 2019. https://doi.org/10.3886/ICPSR36231.v23.

Table 2.  Estimated hazard functions[a] (and 95% confidence intervals[b]) of the age of initiation of hookah use behaviors for PATH¥ young adults (aged 18–24).

| Age | Ever Use (%) | Past 30-Day Use (%) | Fairly Regular Use (%) |
|---|---|---|---|
| 18 | 0.0 | 0.0 | 0.0 |
| 19 | 5.8 (5.0–6.5) | 2.7 (2.2–3.1) | 0.5 (0.3–0.7) |
| 20 | 8.3 (7.2–9.3) | 3.9 (2.4–5.3) | 1.0 (0.4–1.6) |
| 21 | 10.5 (7.6–13.4) | 4.7 (2.5–6.9) | 1.2 (0.7–1.7) |
| 22 | 12.9 (11.6–14.2) | 6.5 (5.5–7.4) | 1.4 (0.8–1.9) |
| 23 | 15.3 (13.8–16.9) | 7.8 (6.6–9.0) | 1.4 (0.9–1.9) |
| 24 | 18.0 (13.6–22.4) | 8.8 (6.6–11.0) | 2.1 (1.4–2.7) |
| 25 | 20.4 (17.3–23.5) | 10.2 (8.5–11.9) | 2.1 (1.4–2.7) |
| 26 | 20.4 (18.4–22.4) | 11.1 (9.5–12.8) | 2.2 (1.6–2.8) |
| 27 | 22.6 (19.9–25.2) | 12.1 (10.0–14.1) | 2.4 (1.7–3.1) |

[a]: Hazard functions are reported as cumulative incidence.

[b]: Turnbull 95% confidence interval.

¥PATH restricted file received disclosure to publish: March 02, 2020 and July 23, 2020. United States Department of Health and Human Services. National Institutes of Health. National Institute on Drug Abuse, and United States Department of Health and Human Services. Food and Drug Administration. Center for Tobacco Products. Population Assessment of Tobacco and Health (PATH) Study [United States] Restricted-Use Files. ICPSR36231-v13. Ann Arbor, MI: Inter-university Consortium for Political and Social Research [distributor], November 5, 2019. https://doi.org/10.3886/ICPSR36231.v23.

initiated fairly regular hookah use. The largest increase in ever and past 30-day hookah use occurred between ages 18 and 19, with 5.8% for ever hookah use and 2.7% for past 30-day hookah use, which represents approximately 1.3 million and over 600 thousand young adults, respectively. Fig 1 presents the overall hazard functions of the age of initiation for all three hookah use behaviors.

Table 3 presents the crude and adjusted hazard ratios estimating the differences in the age of hookah initiation behaviors by sex and by race/ethnicity, while controlling for the total number of other tobacco products ever used. Males were 24% (HR: 1.24, 95% CI: 1.05–1.47) more likely to initiate ever hookah use at an earlier age, 32% more likely (HR: 1.32, 95% CI: 1.06–1.64) to initiate past 30-day hookah use at an earlier age, and 139% more likely (HR:2.39, 95% CI: 1.38–4.15) to initiate fairly regular hookah use at an earlier age compared to females. After controlling for race/ethnicity and the total number of other tobacco products ever used, significant differences by sex were only observed for the age of initiation of fairly regular hookah use (HR: 2.15, 95% CI: 1.22–3.77). Differences by race/ethnicity that were found in the crude models remained after adjustment. After adjusting for sex and the total number of other tobacco products ever used, Hispanic young adults were 59% (HR: 1.59, 95% CI: 1.31–1.94) more likely to initiate ever hookah use at an earlier age, 86% (HR: 1.86, 95% CI: 1.35–2.56) more likely to initiate past 30-day hookah use at an earlier age, and 182% (HR: 2.82, 95% CI: 1.58–5.06) more likely to initiate fairly regular hookah use at an earlier age compared to non-Hispanic White young adults. After adjusting for sex and the total number of other tobacco products ever used, non-Hispanic Black young adults had 102% (HR: 2.02, 95%CI: 1.64–2.48) more likely to initiate ever hookah use at an earlier age and 167% (HR: 2.67, 95% CI: 2.05–3.48) more likely to initiate past 30-day hookah use at an earlier age compared to non-Hispanic White young adults. Finally, after adjusting for sex and race/ethnicity, young adults who have ever used 1 other tobacco product were 53% (HR: 1.53, 95% CI: 1.20–1.94) and 57% (HR: 1.13–2.17, 95% CI: 1.13–2.17) more likely to initiate ever and past 30-day hookah use at earlier

(a) Ever Hookah Use

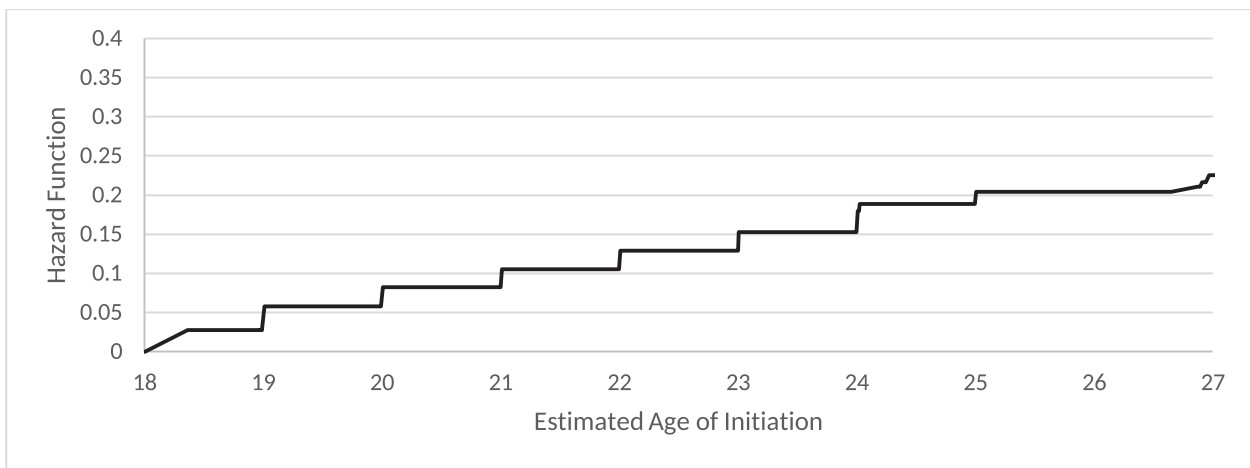

(b) Past 30-Day Hookah Use

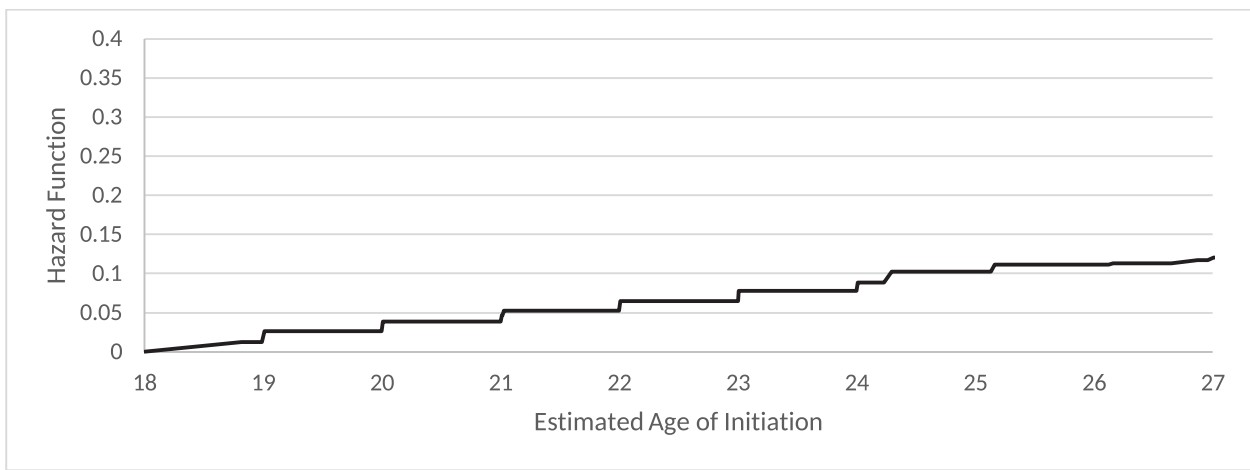

(c) Fairly Regular Hookah Use

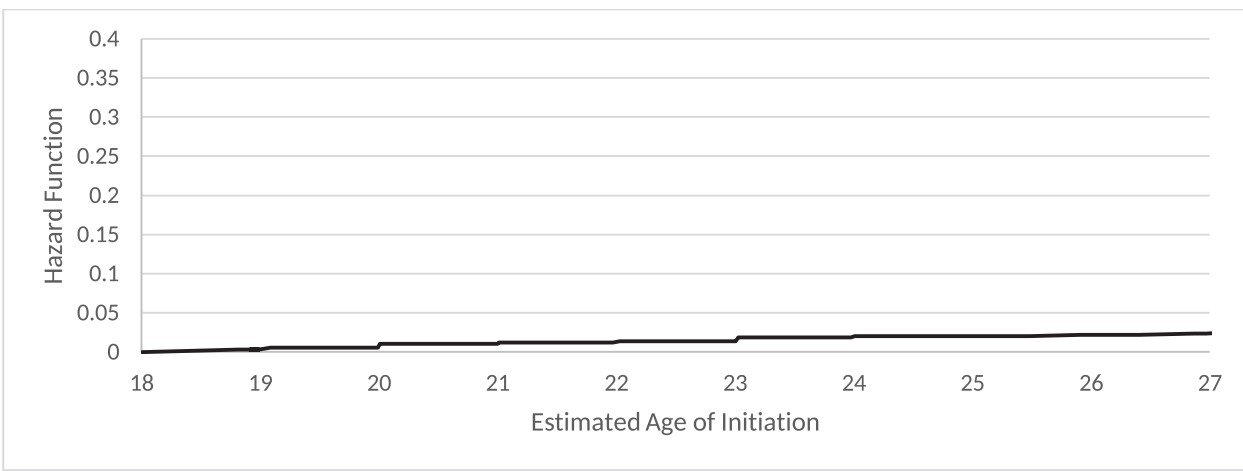

**Fig 1. Estimated hazard function of the age of hookah initiation overall.**

**Table 3. Hazard ratios (and 95% confidence intervals) for each age of initiation of hookah use behaviors by sex, race/ethnicity, and other tobacco products use¥.**

| Variable | Ever Use | Past 30-day Use | Fairly Regular Use |
|---|---|---|---|
| **Univariate Analysis** | | | |
| **Sex** | | | |
| Female | 1.00 | 1.00 | 1.00 |
| Male | **1.24 (1.05–1.47)** | **1.32 (1.06–1.64)** | **2.39 (1.38–4.15)** |
| **Race/Ethnicity** | | | |
| Non-Hispanic White | 1.00 | 1.00 | 1.00 |
| Hispanic | **1.58 (1.29–1.93)** | **1.84 (1.33–2.53)** | **2.80 (1.58–4.98)** |
| Non-Hispanic Black | **1.99 (1.62–2.46)** | **2.62 (2.00–3.44)** | 1.78 (0.90–3.51) |
| Non-Hispanic Other[†] | 1.15 (0.77–1.70) | 1.40 (0.80–2.46) | 1.91 (0.59–6.18) |
| **Number of other tobacco products ever used** | | | |
| 0 tobacco products | 1.00 | 1.00 | 1.00 |
| 1 tobacco product | **1.54 (1.21–1.95)** | **1.59 (1.15–2.19)** | 1.39 (0.64–3.05) |
| 2+ tobacco products | **1.96 (1.63-.234)** | **2.09 (1.68–2.61)** | **2.28 (1.18–4.38)** |
| **Multivariable Analysis** | | | |
| **Sex** | | | |
| Female | 1.00 | 1.00 | 1.00 |
| Male | 1.17 (0.99–1.37) | 1.23 (0.99–1.53) | **2.15 (1.22–3.77)** |
| **Race/Ethnicity** | | | |
| Non-Hispanic White | 1.00 | 1.00 | 1.00 |
| Hispanic | **1.59 (1.31–1.94)** | **1.86 (1.35–2.56)** | **2.82 (1.58–5.06)** |
| Non-Hispanic Black | **2.02 (1.64–2.48)** | **2.67 (2.05–3.48)** | 1.83 (0.93–3.59) |
| Non-Hispanic Other[†] | 1.29 (0.88–1.88) | 1.59 (0.91–2.79) | 2.13 (0.67–6.81) |
| **Number of other tobacco products ever used** | | | |
| 0 tobacco products | 1.00 | 1.00 | 1.00 |
| 1 tobacco product | **1.53 (1.20–1.94)** | **1.57 (1.13–2.17)** | 1.34 (0.62–2.89) |
| 2+ tobacco products | **1.99 (1.66–2.37)** | **2.09 (1.67–2.63)** | **2.22 (1.15–4.30)** |

[†]: Non-Hispanic Other includes Asian, multi-race, and etc.

¥PATH restricted file received disclosure to publish: March 02, 2020 and July 23, 2020. United States Department of Health and Human Services. National Institutes of Health. National Institute on Drug Abuse, and United States Department of Health and Human Services. Food and Drug Administration. Center for Tobacco Products. Population Assessment of Tobacco and Health (PATH) Study [United States] Restricted-Use Files. ICPSR36231-v13. Ann Arbor, MI: Inter-university Consortium for Political and Social Research [distributor], November 5, 2019. https://doi.org/10.3886/ICPSR36231.v23.

ages, respectively, than young adults who had ever used 0 tobacco products. Similarly, after adjustment, young adults who had ever used 2 or more other tobacco products were 99% (HR: 1.99, 95% CI: 1.66–2.37) more likely to initiate ever hookah use at an earlier age, 109% (HR: 2.09, 95% CI: 1.67–2.63) more likely to initiate past 30-day hookah use at an earlier age, and 122% (HR:2.22, 95% CI: 1.15–4.30) more likely to initiate fairly regular hookah use at an earlier age than young adults who had ever used 0 other tobacco products.

Table 4 reports the distribution of the age of initiating hookah use behaviors by sex for U.S. young adults who were never hookah users at their first wave PATH adult participation. By age 21, it is estimated that 11.5% of males and 7.3% of females initiate ever hookah use. The largest increase in ever hookah use occurred between ages 18 and 19 for both males and females (6.7% and 4.9%, respectively). By age 21, 6.4% of males and 4.6% of females initiated past 30-day hookah use. By age 26, the latest age for which we have follow-up for males, 12.4%

**Table 4. Estimated hazard functions[a] (and 95% confidence intervals[b]) of the age of initiation of each hookah use behaviors for the overall sample of PATH¥ young adult (aged 18–24) by sex.**

| Age | Males | Females |
|---|---|---|
| **Initiation of ever hookah use** | | |
| **18** | 0.0 | 0.0 |
| **19** | 6.7 (5.7–7.8) | 4.9 (4.0–5.8) |
| **20** | 9.3 (7.8–10.8) | 7.3 (5.1–9.4) |
| **21** | 11.5 (9.2–13.8) | 7.3 (5.2–9.4) |
| **22** | 14.2 (12.3–16.0) | 11.8 (8.4–15.3) |
| **23** | 16.5 (14.1–19.0) | 14.2 (11.5–17.0) |
| **24** | 20.6 (14.4–26.8) | 17.5 (12.3–22.6) |
| **25** | 20.9 (17.2–24.5) | 18.7 (15.2–22.1) |
| **26** | 22.7 (19.5–25.8) | 18.7 (16.0–21.3) |
| **27** | 24.7 (20.9–28.5) | 20.9 (17.2–24.5) |
| **Initiation of past 30-day hookah use** | | |
| **18** | 0.0 | 0.0 |
| **19** | 3.3 (2.6–4.1) | 2.1 (1.5–2.6) |
| **20** | 4.6 (2.1–7.0) | 3.1 (1.1–5.2) |
| **21** | 6.4 (3.4–9.3) | 4.6 (1.7–7.4) |
| **22** | 7.5 (6.0–8.9) | 5.5 (4.4–6.7) |
| **23** | 8.7 (6.7–10.7) | 6.9 (5.1–8.7) |
| **24** | 10.3 (7.4–13.1) | 9.0 (6.8–11.2) |
| **25** | 11.4 (9.3–13.6) | 9.0 (6.9–11.1) |
| **26** | 12.4 (10.2–14.6) | 10.0 (7.9–12.2) |
| **27** | 12.4 (10.2–14.6) | 11.7 (8.9–14.5) |
| **Initiation of fairly regular hookah use** | | |
| **18** | 0.0 | 0.0 |
| **19** | 0.8 (0.4–1.1) | 0.3 (0.07–0.6) |
| **20** | 0.8 (0.4–1.1) | 0.6 (0.2–0.9) |
| **21** | 1.7 (0.9–2.5) | 0.7 (0.2–1.3) |
| **22** | 1.7 (0.9–2.5) | 1.0 (0.4–1.5) |
| **23** | 2.8 (1.7–3.9) | 1.0 (0.5–1.4) |
| **24** | 2.8 (1.8–3.8) | 1.3 (0.7–1.9) |
| **25** | 3.2 (2.0–4.3) | 1.3 (0.7–1.9) |
| **26** | 3.2 (2.1–4.4) | 1.3 (0.7–1.9) |
| **27** | 3.2 (2.1–4.4) | 1.7 (0.8–2.5) |

[a]: Hazard functions are reported as cumulative incidence.

[b]: Turnbull 95% confidence interval.

[†]: Non-Hispanic Other includes Asian, multi-race, and etc.

¥PATH restricted file received disclosure to publish: March 02, 2020 and July 23, 2020. United States Department of Health and Human Services. National Institutes of Health. National Institute on Drug Abuse, and United States Department of Health and Human Services. Food and Drug Administration. Center for Tobacco Products. Population Assessment of Tobacco and Health (PATH) Study [United States] Restricted-Use Files. ICPSR36231-v13. Ann Arbor, MI: Inter-university Consortium for Political and Social Research [distributor], November 5, 2019. https://doi.org/10.3886/ICPSR36231.v23.

initiated past 30-day hookah use. By age 27, the latest age for which we have follow-up for females, 11.7% initiated past 30-day hookah use. By age 21, it is estimated that 1.7% of males and 0.7% of females initiated fairly regular hookah use. Fig 2 presents the hazard functions of the age of initiation by sex for all three hookah use behaviors.

Table 5 reports the distribution of the age of initiating hookah use behaviors by race/ethnicity for U.S. young adults who were never hookah users at their first wave PATH adult participation. While 8.3% of non-Hispanic White young adults were estimated to initiate ever hookah use by age 21, 9.9%, 11.2%, and 9.1% of Hispanic, non-Hispanic Black, and non-Hispanic other young adults, respectively, were estimated initiate ever hookah use by this age. By age 25, which is considered the upper limit of young adulthood, 15.1% of non-Hispanic White, 23.9% of Hispanic, 32.4% non-Hispanic Black, and 17.8% of non-Hispanic other young adults, were estimated to initiate ever hookah use. By age 21, 2.7% of non-Hispanic White, 5.6% of Hispanic, 6.1% of non-Hispanic Black, and 5.1% of non-Hispanic other young adults initiated past 30-day hookah use. By age 21, 0.8% of non-Hispanic White, 1.7% of Hispanic, 0.9% of non-Hispanic Black, and 1.7% of non-Hispanic other young adults initiated fairly regular hookah use. Fig 3 presents the hazard functions of the age of initiation by race/ethnicity for all three hookah use behaviors.

Table 6 reports the hazard ratios of the age of initiating hookah use behaviors by prior use of six other tobacco products. Univariate analyses revealed that prior use of cigarettes, e-cigarettes, traditional cigars, cigarillos, filtered cigars, and smokeless tobacco were associated with an earlier age of ever hookah initiation. In addition, prior use of cigarettes, e-cigarettes, cigarillos, filtered cigars, and smokeless tobacco also increased the risk of an earlier age of past 30-day hookah initiation. Finally, prior use of e-cigarettes, traditional cigars, cigarillos, filtered cigars, and smokeless tobacco all increased the risk of an earlier age of fairly regular hookah initiation. After controlling for sex and race/ethnicity, prior use cigarillos increased the risk of an earlier age of (i) ever hookah initiation, (ii) past 30-day hookah initiation, and (iii) fairly regular hookah initiation. After controlling for sex and race/ethnicity, prior use of e-cigarettes increased the risk for an earlier age of ever hookah initiation (AHR = 1.42; 95%CI = 1.15–1.75), while prior use of cigarettes increased the risk for an earlier age of past 30-day hookah initiation (AHR = 1.35; 95%CI = 1.00–1.82).

## Discussion

To the best of our knowledge, our study is the first in the field to prospectively estimate the distribution of the age of initiation of ever hookah use, past 30-day hookah use, and fairly regular hookah use among young adult never hookah users. Our results are consistent with findings from the 2010–2011 Current Population Survey of hookah use by young adults (18–25 years old) that reported that 5.6% had ever used hookah and 1.1% had used hookah in the past 30-days [22] and the 2015 National Health Interview Survey found that 3.4% of young adults (18–24 years old) reported using hookah every day or some days at the time of survey [29]. Our findings are also similar to a different PATH study that included never users of hookah in 2013–2014 and were followed-up in 2014–2015, which found that 9.8% had initiated ever hookah and 5.0% initiated past 30-day hookah use one year later [30]. In addition, a different PATH study reported that among young adult (18–24 years old) never users of hookah at each wave (waves 1–2, 2013–2015), 14.1% initiated ever use, 6.1% initiated past 30-day use, and 0.2% initiated frequent hookah use (hookah use on at least 20 days in the past 30-days) one year later [18]. However, our study extends these previous PATH studies by estimating the age at which young adults initiate different hookah use behaviors and by including never hookah users at waves 2 and 3. Importantly, in our study, between ages 21 to 25, the estimated percentage of users almost doubled for all three hookah use behaviors.

Previous studies of hookah use by sex have reported mixed results [19,31]. For example, a previous report of the 2013–2014 PATH study young adults (18–24 years old) found that the prevalence of ever hookah use among males was 41.5% and 46.8% among females [31]. This

(a) Ever Hookah Use

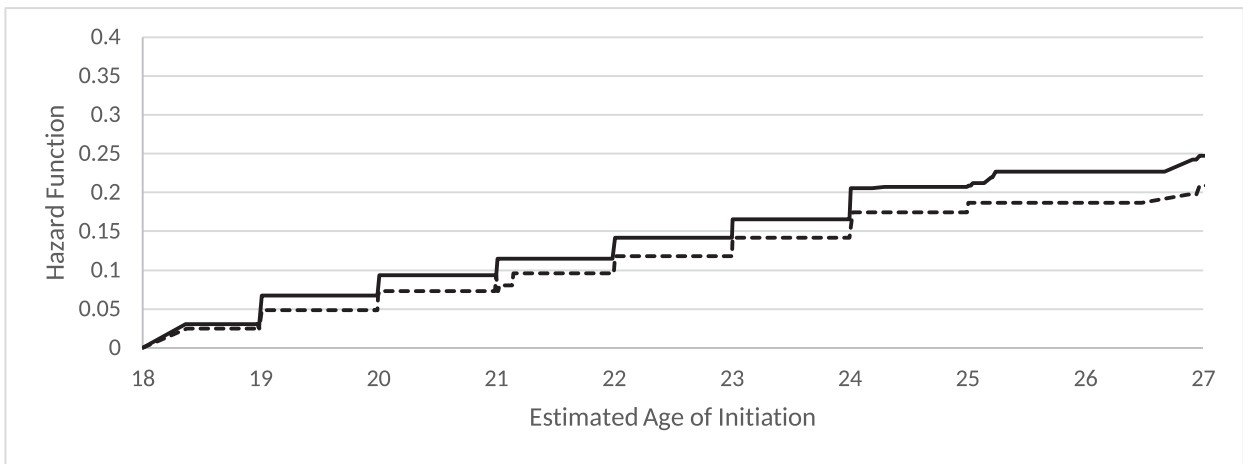

(b) Past 30-Day Hookah Use

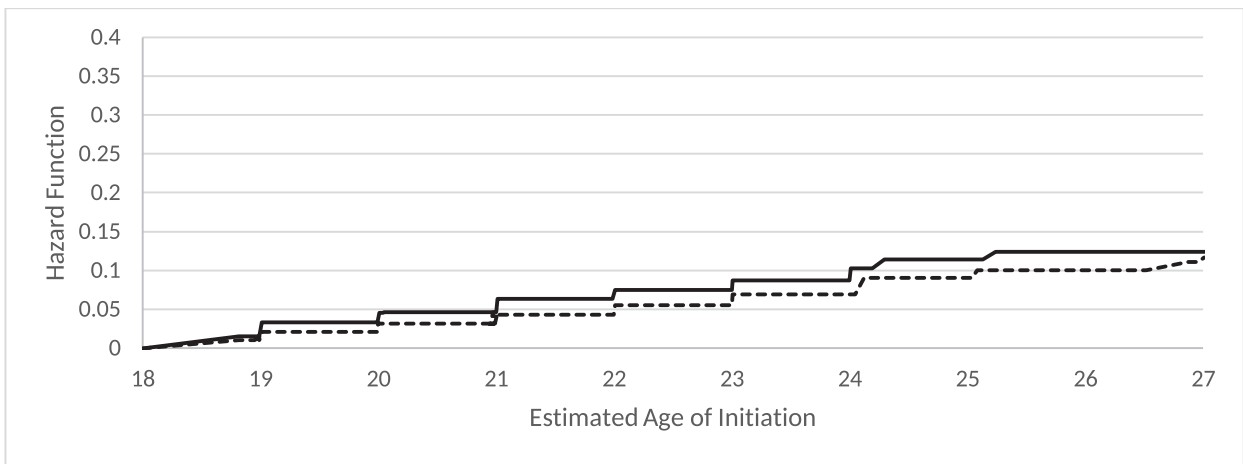

(c) Fairly Regular Hookah Use

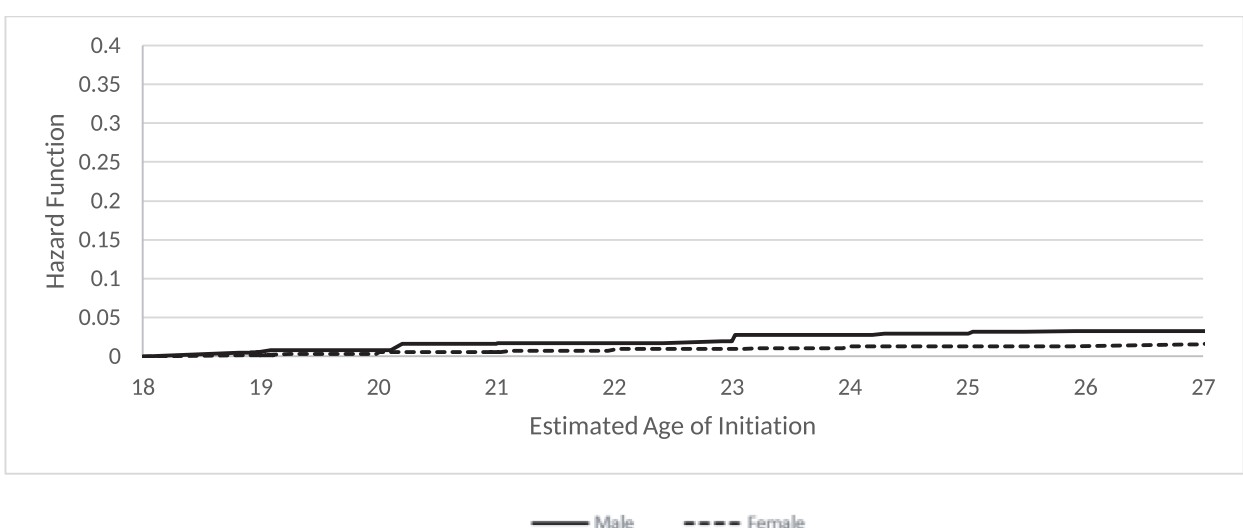

**Fig 2. Estimated hazard function of the age of hookah initiation by sex.**

**Table 5. Estimated hazard functions[a] (and 95% confidence intervals[b]) for the age of hookah initiation behaviors for the PATH¥ young adults (aged 18–24) by race/ethnicity [†].**

| Age | Non-Hispanic White | Hispanic | Non-Hispanic Black | Non-Hispanic other[†] |
|---|---|---|---|---|
| **Initiation of ever hookah use** | | | | |
| 18 | 0.0 | 0.0 | 0.0 | 0.0 |
| 19 | 5.2 (4.2–6.3) | 6.9 (5.4–8.4) | 6.9 (5.1–8.7) | 5.0 (2.8–7.1) |
| 20 | 6.9 (5.6–8.2) | 9.9 (7.2–12.7) | 11.2 (5.2–17.3) | 7.5 (4.2–10.8) |
| 21 | 8.3 (5.4–11.2) | 9.9 (6.7–13.2) | 11.2 (8.7–13.7) | 9.1 (4.5–13.6) |
| 22 | 8.7 (7.3–10.1) | 15.8 (13.6–18.1) | 18.8 (15.0–22.7) | 11.0 (7.2–14.7) |
| 23 | 11.7 (9.2–14.3) | 18.4 (14.6–22.3) | 23.0 (16.7–29.2) | 14.3 (8.4–20.3) |
| 24 | 14.3 (10.1–18.5) | 20.8 (15.4–26.1) | 23.0 (19.5–26.5) | 17.8 (9.5–26.2) |
| 25 | 15.1 (12.4–17.8) | 23.9 (19.7–28.1) | 32.4 (24.6–40.1) | 17.8 (10.2–25.5) |
| 26 | 15.6 (12.8–18.4) | 25.4 (20.7–30.1) | 32.7 (27.7–37.7) | 18.7 (11.0–26.4) |
| 27 | 15.8 (12.8–18.8) | 26.8 (21.6–32.0) | 40.1 (30.7–49.5) | 19.3 (11.4–27.3) |
| **Initiation of past 30-day hookah use** | | | | |
| 18 | 0.0 | 0.0 | 0.0 | 0.0 |
| 19 | 2.0 (1.4–2.6) | 3.9 (1.6–6.2) | 3.6 (2.3–5.0) | 2.3 (1.2–3.3) |
| 20 | 2.0 (0.0–4.0) | 5.6 (2.5–8.8) | 6.1 (4.6–7.6) | 3.4 (1.5–5.2) |
| 21 | 2.7 (0.2–5.2) | 5.6 (4.0–7.3) | 6.1 (2.4–9.8) | 5.1 (1.4–8.7) |
| 22 | 4.4 (3.5–5.4) | 8.4 (6.3–10.6) | 11.2 (7.7–14.6) | 6.0 (3.4–8.5) |
| 23 | 4.4 (3.2–5.7) | 9.8 (6.4–13.2) | 13.5 (10.7–16.2) | 8.0 (4.4–11.5) |
| 24 | 5.4 (3.5–7.3) | 11.4 (7.7–15.1) | 13.5 (10.8–16.1) | 8.0 (4.6–11.4) |
| 25 | 6.8 (5.0–8.7) | 12.9 (9.2–16.5) | 19.3 (15.5–23.1) | 10.8 (3.1–18.4) |
| 26 | 7.7 (5.5–9.8) | 12.9 (9.7–16.1) | 21.0 (16.8–25.3) | 11.3 (3.6–19.1) |
| 27 | 8.4 (5.9–11.0) | 12.9 (9.8–16.0) | 24.3 (18.5–30.2) | 12.5 (4.3–20.7) |
| **Initiation of fairly regular hookah use** | | | | |
| 18 | 0.0 | 0.0 | 0.0 | 0.0 |
| 19 | 0.6 (0.2–1.0) | 0.5 (0.06–1.0) | 0.2 (0.0–0.5) | 0.3 (0.0–0.7) |
| 20 | 0.7 (0.3–1.1) | 1.7 (0.7–2.7) | 0.7 (0.03–1.4) | 1.6 (0.0–3.9) |
| 21 | 0.8 (0.3–1.4) | 1.7 (0.7–2.7) | 0.9 (0.3–1.6) | 1.7 (0.0–3.6) |
| 22 | 0.9 (0.5–1.4) | 2.4 (1.3–3.4) | 0.9 (0.0–2.0) | 1.7 (0.0–3.8) |
| 23 | 0.9 (0.5–1.4) | 2.4 (1.3–3.4) | 2.5 (0.2–4.8) | 2.4 (0.0–4.8) |
| 24 | 1.2 (0.6–1.8) | 3.6 (2.2–5.1) | 2.6 (1.1–4.0) | 2.4 (0.0–4.8) |
| 25 | 1.2 (0.6–1.8) | 4.4 (2.7–6.1) | 2.7 (1.2–4.2) | 2.4 (0.0–4.8) |
| 26 | 1.2 (0.6–1.8) | 4.4 (2.7–6.1) | 2.7 (1.2–4.2) | 2.4 (0.0–4.8) |
| 27 | N/A | 4.4 (2.7–6.1) | 4.3 (1.5–7.1) | 2.9 (0.2–5.7) |

[a]: Hazards are cumulative incidences.

[b]: Turnbull 95% confidence interval.

[†]: Non-Hispanic other includes Asian, multi-race, and etc.

¥PATH restricted file received disclosure to publish: March 02, 2020 and July 23, 2020. United States Department of Health and Human Services. National Institutes of Health. National Institute on Drug Abuse, and United States Department of Health and Human Services. Food and Drug Administration. Center for Tobacco Products. Population Assessment of Tobacco and Health (PATH) Study [United States] Restricted-Use Files. ICPSR36231-v13. Ann Arbor, MI: Inter-university Consortium for Political and Social Research [distributor], November 5, 2019. https://doi.org/10.3886/ICPSR36231.v23.

study also found that the prevalence of past 30-day hookah use among young adults was 9.0% in males and 12.4% in females [31]. These results are different from our study, which found that males have increased risk of initiating hookah use at earlier ages compared to females. The difference in these results could be due to the fact that the 2013–2014 study of PATH included hookah users, while our study started with never users, measured hookah initiation at follow-

(a) Ever Hookah Use

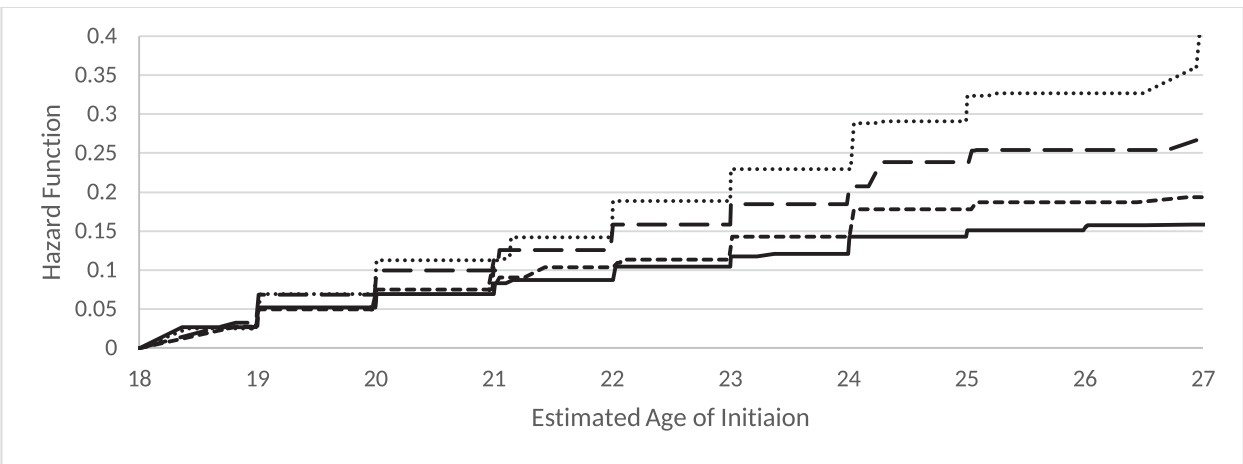

(b) Past 30-Day Hookah Use

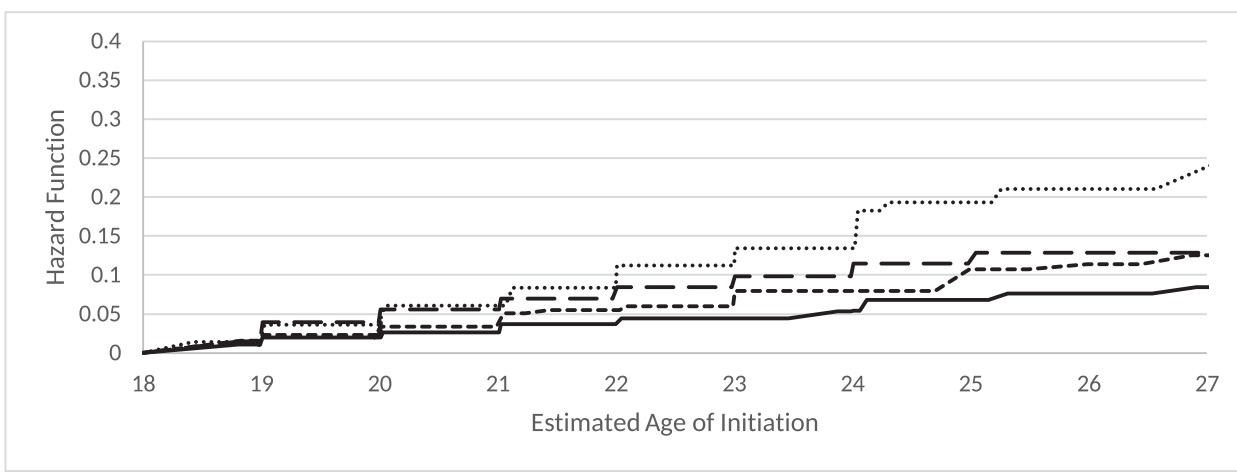

(c) Fairly Regular Hookah Use

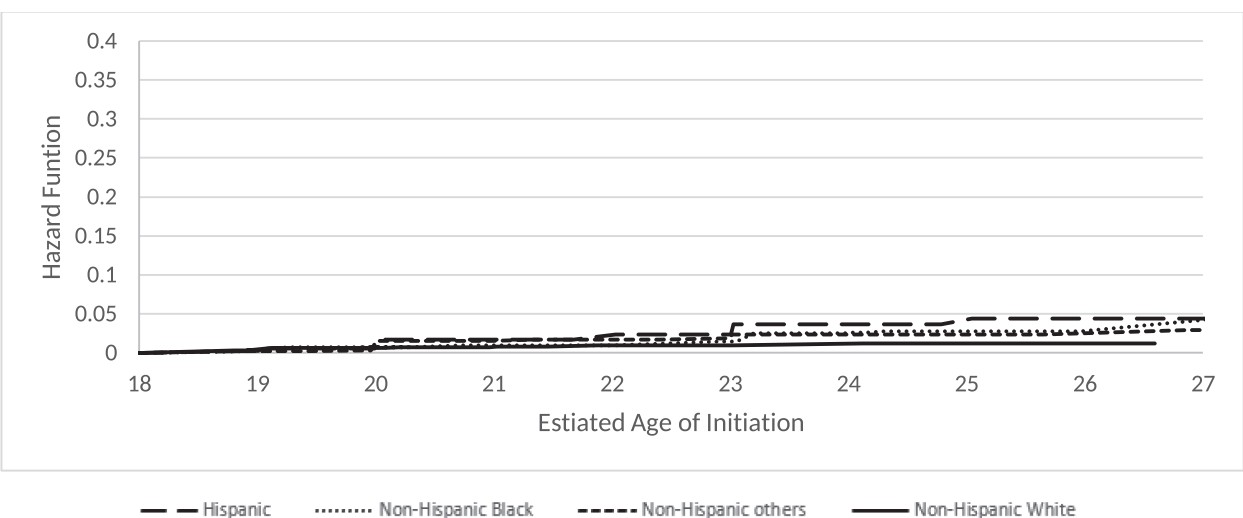

**Fig 3. Estimated hazard function of the age of hookah initiation by race/ethnicity.**

**Table 6. Association of prior ever other tobacco product use with the age of initiation of hookah use behaviors for the PATH¥ young adults (aged 18–24).**

| | | Ever Use | Past 30-day Use | Fairly Regular Use |
|---|---|---|---|---|
| **Crude hazard ratios** | | | | |
| **Prior Ever use of Other Tobacco Product Before Hookah Initiation** | | | | |
| Cigarette | No | 1.00 | 1.00 | 1.00 |
| | Yes | **1.70 (1.45–2.01)** | **1.90 (1.54–2.35)** | 1.66 (0.94–2.96) |
| E-cigarette | No | 1.00 | 1.00 | 1.00 |
| | Yes | **1.82 (1.55–2.13)** | **1.73 (1.40–2.13)** | **2.64 (1.50–4.66)** |
| Smokeless Tobacco | No | 1.00 | 1.00 | 1.00 |
| | Yes | **1.62 (1.29–2.04)** | **1.64 (1.17–2.31)** | **2.34 (1.16–4.74)** |
| Traditional Cigars | No | 1.00 | 1.00 | 1.00 |
| | Yes | **1.46 (1.16–1.84)** | 1.21 (0.85–1.72) | **1.86 (1.02–3.40)** |
| Filtered Cigars | No | 1.00 | 1.00 | 1.00 |
| | Yes | **1.66 (1.34–2.04)** | **1.90 (1.39–2.60)** | **2.85 (1.52–5.35)** |
| Cigarillo | No | 1.00 | 1.00 | 1.00 |
| | Yes | **1.97 (1.66–2.34)** | **2.22 (1.77–2.79)** | **2.86 (1.68–4.88)** |
| **Adjusted hazard ratios[a]** | | | | |
| **Prior Ever use of Other Tobacco Product Before Hookah Initiation** | | | | |
| Cigarette | No | 1.00 | 1.00 | 1.00 |
| | Yes | 1.17 (0.96–1.44) | **1.35 (1.00–1.82)** | 0.57 (0.28–1.12) |
| E-cigarette | No | 1.00 | 1.00 | 1.00 |
| | Yes | **1.42 (1.15–1.75)** | 1.20 (0.91–1.60) | 2.13 (0.84–5.38) |
| Smokeless Tobacco | No | 1.00 | 1.00 | 1.00 |
| | Yes | 1.25 (0.96–1.61) | 1.32 (0.89–1.96) | 1.32 (0.66–2.61) |
| Traditional Cigars | No | 1.00 | 1.00 | 1.00 |
| | Yes | 0.98 (0.74–1.30) | 0.70 (0.46–1.06) | 0.80 (0.33–1.95) |
| Filtered Cigars | No | 1.00 | 1.00 | 1.00 |
| | Yes | 1.01 (0.79–1.29) | 1.25 (0.88–1.78) | 1.60 (0.79–3.23) |
| Cigarillo | No | 1.00 | 1.00 | 1.00 |
| | Yes | **1.37 (1.06–1.76)** | **1.54 (1.08–2.21)** | **2.35 (1.22–4.52)** |

[a]: Hazard ratios (and 95% confidence intervals) are controlled for sex and race/ethnicity.

¥PATH restricted file received disclosure to publish: March 29, 2021. United States Department of Health and Human Services. National Institutes of Health. National Institute on Drug Abuse, and United States Department of Health and Human Services. Food and Drug Administration. Center for Tobacco Products. Population Assessment of Tobacco and Health (PATH) Study [United States] Restricted-Use Files. ICPSR36231-v13. Ann Arbor, MI: Inter-university Consortium for Political and Social Research [distributor], November 5, 2019. https://doi.org/10.3886/ICPSR36231.v23.

up waves, and estimated the age of initiation. We found no other recent studies that have examined hookah use by sex to compare our results with.

Previous studies of hookah use by race/ethnicity have also reported mixed results. In contrast to our findings, a study using the Current Population Survey of 18–30 year olds showed that the combined prevalence of ever hookah use from the years 2010–2011 and 2014–2015 was the lowest among non-Hispanic Black (4.6%), 7.0% among Hispanic, and 11.8% among non-Hispanic White [32], which is different from our findings that non-Hispanic Black and Hispanic young adults had increased risk of initiating hookah use at earlier ages compared to non-Hispanic White young adults. We also note that the prevalence of hookah use is much lower for each race/ethnicity group by age 26–27 (the latest age for which we had follow-up)

compared to our study. One potential explanation is that the Current Population Survey includes 28–30 year olds who may have a high prevalence of never hookah use compared to our study. A 2009 study of college students at the University of Florida reported that there were differences in hookah by race, as White students had increased odds of ever hookah use compared to Black students (AOR = 5.0; 95%CI = 3.0–8.5) [21], which is different from our findings that non-Hispanic Black young adults had increased risk of initiating hookah use at earlier ages. However, this study was only conducted among college students in Florida, while our study is a nationally representative sample of young adults. A similar study was conducted among youth (12–17 years old) never hookah users at their first wave of PATH participation, which also found that Hispanic youth had increased risk of initiating ever hookah use at earlier ages compared to non-Hispanic White youth [33]. However this study found there was no difference in the age of hookah initiation between non-Hispanic Black youth and non-Hispanic White youth [33], which was observed among the young adults in our study.

A different study of the 2013–2014 PATH young adults found that 10.0% non-Hispanic White, 12.8% of Hispanic, 10.2% of non-Hispanic Black,10.2% of non-Hispanic Asian, and 11.9% of non-Hispanic other reported past 30-day hookah use [31]. A previous cross-sectional study in 2008–2009 which found that the prevalence of past 30-day hookah use was 3.9% among non-Hispanic Black, 9.9% among non-Hispanic White, and 10.3% among Hispanic college students 18–24 years old [19], which is different from our findings that non-Hispanic Black and Hispanic young adults have increased risk of initiating hookah at earlier ages. The differences between these findings could be attributed to the previous study being conducted among Florida college students, while our study was conducted among a nationally representative sample of young adults. Our findings that Non-Hispanic Black and Hispanic young adults have earlier ages of hookah initiation compared to Non-Hispanic White young adults indicates that these minority groups could benefit from culturally-relevant targeted hookah intervention programs.

Additionally, given that prior cigarillo and e-cigarette use increased the risk of an earlier age of hookah initiation, more research is needed to determine if the use of flavored tobacco products reinforces the use of hookah, which is also a flavored tobacco product. A previous study in 2013–2014 among young adults showed that cigarette use predicts ever hookah initiation 6 months later [34], which is consistent with our findings. A PATH study from 2013–2016 among adults showed that e-cigarette and cigar product use predicted past 30-day hookah use after controlling for other covariates [24], which we confirmed with our findings among young adults on the age of initiation of hookah use.

A previous study of PATH waves 1–2 (2013–2015) estimating the association between perceptions of the harmfulness of hookah compared to cigarettes with initiation of ever hookah use at wave 2 found that those who perceive hookah as less harmful had increased odds of initiating hookah use (HR = 2.89; 95%CI = 1.82–4.61) compared to those who perceive hookah as more harmful than cigarettes [35]. Interestingly, recent studies suggest that most hookah users in college do not admit to their health care providers that they use hookah [36], potentially because of the infrequent use patterns of hookah products [37] and they incorrectly think that they are free from the risk of nicotine addiction [38]. Therefore, education and intervention campaigns are needed to correct misperceptions about the harmfulness and addictiveness of hookah among young adults, as our results show that many young adults initiate ever or past 30-day hookah users in this developmental stage of life. Prior studies have shown that young adults report significantly greater perceived harm and addictiveness about hookah after they received information/messages about the risks associated with use of the product [39,40], and that these messages can change the young adults' attitude toward hookah use [40]. Thus, it may be helpful for tobacco regulatory entities to develop mass media campaigns with messages

about the risks of hookah use [41]. Correcting the misperceptions about hookah use is needed because hookah use exposes users to chemicals inducing nicotine dependence [2,3,5,6] and has the potential to result in adverse health outcomes. By identifying the age at which young adults initiate hookah behaviors, intervention and prevention campaigns can use this information to educate young adults on the potential harms of hookah use before they start using hookah or before they progress to more frequent hookah use. Our results suggest that the hazard function of hookah use doubled between ages 21–25, which suggests that prevention and intervention campaigns should target young adults between 18–21 years old to prevent hookah initiation.

## Strength and limitations

We utilized a nationally representative study to prospectively estimate the age of initiation of ever, past 30-day, and fairly regular hookah use, which is one of the strengths of our study. Another strength is the longitudinal design of our study, which allowed us to follow our sample of young adults over multiple years (2013–2017) to prospectively estimate the age of hookah initiation behaviors, which is not subject to recall bias. Interval censoring with non-parametric methods to estimate the age of initiation is another strength, as our results do not depend on parametric model assumptions. It should be noted that the measure for "fairly regular" hookah use is a subjective measure, which does not allow us to quantify the amount of hookah use and can be considered a limitation. In order to protect participant confidentiality, PATH participant birth dates are not included in the restricted-use data, which prevented us from obtaining participants' exact age at each wave, and is a limitation. We circumvent this limitation by using the number of weeks between survey waves and interval-censoring to estimate the age of initiation.

## Conclusion

In conclusion, our study examined the age of initiation of hookah use among young adults in the United States. We provide evidence to the field on the prospective estimates of the age of initiation of ever, past 30-day, and fairly regular hookah use, as well as differences by sex, race/ethnicity. Finally, we examined the association of prior use of six other tobacco products on the age of initiation of ever, past 30-day, and fairly regular hookah use. Considering that young adulthood is a developmental phase where patterns of tobacco use become less volatile and more stable [42], prevention and intervention campaigns are needed urgently especially for Hispanic young adults, Non-Hispanic Black young adults, and male young adults, as these groups are particularly vulnerable to earlier onset of hookah use.

## Supporting information

**S1 Table. Frequency distribution of ever use of other tobacco products prior to past 30-day hookah use and fairly regular hookah use.**
(DOCX)

## Author Contributions

**Conceptualization:** Adriana Pérez, Meagan A. Bluestein.

**Data curation:** Arnold E. Kuk.

**Formal analysis:** Adriana Pérez, Arnold E. Kuk.

**Funding acquisition:** Adriana Pérez, Baojiang Chen, Melissa B. Harrell.

**Investigation:** Adriana Pérez, Arnold E. Kuk.

**Methodology:** Adriana Pérez, Baojiang Chen.

**Project administration:** Adriana Pérez.

**Resources:** Arnold E. Kuk, Meagan A. Bluestein.

**Software:** Adriana Pérez.

**Supervision:** Adriana Pérez.

**Validation:** Meagan A. Bluestein.

**Visualization:** Adriana Pérez, Arnold E. Kuk, Meagan A. Bluestein.

**Writing – original draft:** Adriana Pérez, Arnold E. Kuk.

**Writing – review & editing:** Adriana Pérez, Arnold E. Kuk, Meagan A. Bluestein, Baojiang Chen, Kymberle L. Sterling, Melissa B. Harrell.

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
