## [Decision Letter · Decision Letter 0]

11 Aug 2021

PONE-D-21-11766

Age of initiation of hookah use among young adults: Findings from the Population Assessment of Tobacco and Health (PATH) study, 2013-2017

PLOS ONE

Dear Dr. Pérez,

Thank you for submitting your manuscript to PLOS ONE. After careful consideration, we feel that it has merit but does not fully meet PLOS ONE’s publication criteria as it currently stands. Therefore, we invite you to submit a revised version of the manuscript that addresses the points raised during the review process.

We look forward to receiving your revised manuscript.

Kind regards,

Maria R. Khan, PhD, MPH

Academic Editor

PLOS ONE

1. Please ensure that your manuscript meets PLOS ONE's style requirements, including those for file naming. The PLOS ONE style templates can be found at https://journals.plos.org/plosone/s/file?id=wjVg/PLOSOne_formatting_sample_main_body.pdf and https://journals.plos.org/plosone/s/file?id=ba62/PLOSOne_formatting_sample_title_authors_affiliations.pdf.

2. If materials, methods, and protocols are well established, authors may cite articles where those protocols are described in detail, but your submission should include sufficient information to be understood independent of these references (https://journals.plos.org/plosone/s/submission-guidelines#loc-materials-and-methods). As such, please ensure that your details on the PATH study are sufficiently well-described external to any previous literature on this research.

Additional Editor Comments (if provided):

Please respond to the reviewer concerns and in particular make clear how these findings address a gap in the literature.

Reviewers' comments:

Reviewer's Responses to Questions

**Comments to the Author**

1. Is the manuscript technically sound, and do the data support the conclusions?

Reviewer #1: Yes

2. Has the statistical analysis been performed appropriately and rigorously? 

Reviewer #1: Yes

3. Have the authors made all data underlying the findings in their manuscript fully available?

Reviewer #1: Yes

4. Is the manuscript presented in an intelligible fashion and written in standard English?

Reviewer #1: Yes

5. Review Comments to the Author

Reviewer #1: The authors submit a retrospective study using restricted use files from PATH to analyze the association between age and intensity of hookah use initiation among different demographic groups as a step toward understanding where to target future education interventions. This is a worthwhile study, the manuscript is well written, and the analysis seems robust. However, I have concerns about the novelty of the results given how much has been published on the topic to date.

Major comments:

- Data for youth (<18 years old) exists in PATH so I am curious why age 18 was chosen at the cut off. My assumption is that children initiate hookah use prior to age 18 and earlier targeted intervention may be even more beneficial. Why was this age group not included?

- For the other tobacco product variable –your data suggest a differential influence of certain other tobacco products (Table 6) on your outcome. I can’t seem to find how and where this variable was used and if it accounts for that. Accordingly, was this variable treated as ordinal or categorical in whichever models it was included?

Minor comments:

- Given the somewhat vague and subjective wording regarding “fairly regular” use, perhaps some mention of this as a limitation to understanding true intensity of use would be beneficial in the discussion

- I’m confused about how this study “prospectively estimates” initiation of hookah use. I realize your results can be predictive or prognostic but the word prospective seems incorrect.

- The discussion would be improved with some added insight into how and why your results differ from prior studies and some additional framing of the importance of your results.

- Similarly, why do you believe there differences between this study’s results and prior studies that have also used PATH data?

6. PLOS authors have the option to publish the peer review history of their article (what does this mean?). If published, this will include your full peer review and any attached files.

Reviewer #1: No

---

## [Editor Report · Decision Letter 1]

28 Sep 2021

Age of initiation of hookah use among young adults: Findings from the Population Assessment of Tobacco and Health (PATH) study, 2013-2017

PONE-D-21-11766R1

Dear Dr. Perez,

We’re pleased to inform you that your manuscript has been judged scientifically suitable for publication and will be formally accepted for publication once it meets all outstanding technical requirements.

Kind regards,

Maria R. Khan, PhD, MPH

Academic Editor

PLOS ONE
---

## [Editor Report · Acceptance letter]

4 Oct 2021

PONE-D-21-11766R1 

Age of initiation of hookah use among young adults: Findings from the Population Assessment of Tobacco and Health (PATH) study, 2013-2017 

Dear Dr. Pérez:

I'm pleased to inform you that your manuscript has been deemed suitable for publication in PLOS ONE. Congratulations! Your manuscript is now with our production department. 

Kind regards, 

on behalf of

Dr. Maria R. Khan 

Academic Editor

PLOS ONE